# Overcoming Order in Autoregressive Graph Generation for Molecule Generation

**Edo Cohen-Karlik**                                    *edocohen@mail.tau.ac.il*
*Verily Research*
*School of Computer Science, Tel Aviv University*

**Eyal Rozenberg**                                     *eyalrozenberg@google.com*
*Verily Research*

**Daniel Freedman**                                    *danielfreedman@google.com*
*Verily Research*

**Reviewed on OpenReview:** *https://openreview.net/forum?id=BK6Gc1OtRy*

## Abstract

Graph generation is a fundamental problem in various domains, and is of particular interest in chemistry where graphs may be used to represent molecules. Recent work has shown that molecular graph generation using recurrent neural networks (RNNs) is advantageous compared to traditional generative approaches which require converting continuous latent representations into graphs. One issue which arises when treating graph generation as sequential generation is the arbitrary order of the sequence which results from a particular choice of graph flattening method: in the chemistry setting, molecular graphs commonly have multiple SMILES strings corresponding to the same molecule. Inspired by the use case of molecular graph generation, we propose using RNNs, taking into account the non-sequential nature of graphs by adding an Orderless Regularization (OLR) term that encourages the hidden state of the recurrent model to be invariant to different valid orderings present under the training distribution. We demonstrate that sequential molecular graph generation models benefit from our proposed regularization scheme, especially when data is scarce. Our findings contribute to the growing body of research on graph generation and provide a valuable tool for various applications requiring the synthesis of realistic and diverse graph structures.

## 1 Introduction

Graphs are powerful representations of complex relationships and structures. While graphs have many application domains, we are particularly interested in their use in chemistry, in which setting graphs may be used to represent molecules. A dedicated class of architectures, Graph Neural Networks (GNNs), has been developed to handle the specific properties of graphs. Graphs are naturally versatile objects, but such versatility comes at the cost of lack of structure and no naturally induced order. Most GNN architectures therefore operate by applying a neural architecture at the node level followed by an aggregation step which takes into account the local neighborhood structure of the graph. By stacking multiple such layers, a GNN is able to perform node-level or graph-level tasks that take into account the entire structure of the graph.

The ability to generate realistic and structured graphs is essential for various applications ranging from drug design (De Cao & Kipf, 2018; Du et al., 2022b; Honda et al., 2019; Jin et al., 2018; Madhawa et al., 2019; Shi et al., 2020; You et al., 2018a; Zang & Wang, 2020) to program synthesis (Brockschmidt et al., 2018; Hindle et al., 2016; Chen et al., 2021a; Allamanis et al., 2017; Hellendoorn et al., 2019; Yin & Neubig, 2017; Bielik et al., 2016; Dai et al., 2018). In recent years a wide variety of generative models have been developed,

including generative adversarial networks (GANs), variational autoencoders (VAEs), normalizing flows, and diffusion models. These algorithms devise different strategies to learn continuous mappings from a latent distribution to a space of realistic examples. Unfortunately, graphs do not admit a natural representation in a continuous space; consequently, the discrete and unordered nature of graphs make them less amenable to the methods mentioned above for the task of graph generation. A different type of generative model relies on autoregressive architectures which enable processing a sequence and generating the next element; for example, these architectures are commonly used for large language models. Generally, autoregressive models are applicable when the generated objects admit a sequential order.

In this work we focus on sequential generation of graphs using autoregressive neural architectures. A strong motivating factor for choosing autoregressive architectures is that we are particularly interested in molecular graph generation; and in this context Flam-Shepherd et al. (2022) have shown that sequential generation is favorable compared to other approaches. The specific representation we consider is depth-first search (DFS) trajectories of graphs. The reasons for this choice of representation is twofold: (a) DFS is a natural way of flattening graphs into sequences; (b) in the chemistry community DFS is used to convert molecules into strings. However, an issue arises when converting graphs into sequences: there are many DFS trajectories for a given graph. Indeed, for many graph flattening methods, there is an arbitrariness in the order of the sequence which results (Vinyals et al., 2015; Chen et al., 2021b).

In order to alleviate the dependency on a specific trajectory, we add a regularization term dubbed Orderless Regularization (OLR) which ensures the learnt model is invariant to different DFS orderings of the same graph. For the sake of training with OLR, one needs to generate different DFS trajectories with a common end-vertex which is known to be hard (Beisegel et al., 2019). While our motivation originates from the use-case of drug discovery and molecular graph generation, we provide a general formalism of the notion of graph-level invariance and devise an efficient algorithm to generate common end-vertex trajectories under certain constraints. Finally, we demonstrate empirically that our regularization term is beneficial when the amount of training data is limited by considering the use case of small molecule generation.

The reminder of the paper is structured as follows: in Section 2 we provide background and introduce the concepts, definitions, and notations used throughout the paper. Section 3 goes into the details of OLR over DFS trajectories. Section 4 is devoted to related work. In Section 5 we provide empirical evidence for the effectiveness of OLR. Section 7 provides concluding remarks.

## 2 Background

In this section we formally define the problem of graph generation, the notations and definitions necessary to present our proposed method. We denote matrices by bold uppercase letters, $\mathbf{M} \in \mathbb{R}^{n \times m}$, vectors by bold lowercase letters, $\boldsymbol{v}$, and the $i^{th}$ entry of $\boldsymbol{v}$ by $v_i$. We proceed with a general formulation of recurrent models.

### 2.1 Recurrent Models

Let $\mathcal{X}$, $\mathcal{H}$, and $\mathcal{Y}$ be the spaces of inputs, hidden states, and outputs, respectively. Given an input sequence $\mathbf{x} \equiv (\boldsymbol{x}_1, \ldots, \boldsymbol{x}_n) \in \mathcal{X}^n$, a recurrent model consists of two functions, the state update function $f_u : \mathcal{H} \times \mathcal{X} \to \mathcal{H}$,

$$\boldsymbol{h}_{t+1} = f_u(\boldsymbol{h}_t, \boldsymbol{x}_t), \tag{2.1}$$

and the output function $f_o : \mathcal{H} \times \mathcal{X} \to \mathcal{Y}$,

$$\boldsymbol{y}_t = f_o(\boldsymbol{h}_t, \boldsymbol{x}_t). \tag{2.2}$$

where $\boldsymbol{h}_0 \in \mathcal{H}$. We overload the notation and denote the hidden state and output of a recurrent model over a sequence as $f_u(\mathbf{x})$ and $f_o(\mathbf{x})$ respectively.

The formulation presented of recurrent models is broad and able to capture RNNs as well as more complex architectures such as Gated Recurrent Units (GRUs, (Chung et al., 2014)) and Long-Short Term Memory networks (LSTMs, (Hochreiter & Schmidhuber, 1997)). For example, in the case of a very simple, vanilla

RNN,[1]

$$\boldsymbol{h}_{t+1} = \sigma_h(\mathbf{A}\boldsymbol{h}_t + \mathbf{B}\boldsymbol{x}_t) \tag{2.3}$$

and

$$\boldsymbol{y}_t = \sigma_y(\mathbf{C}\boldsymbol{h}_t + \mathbf{D}\boldsymbol{x}_t) \tag{2.4}$$

where $\mathbf{A}, \mathbf{B}, \mathbf{C}, \mathbf{D}$ are matrices with the appropriate dimensions and $\sigma_h, \sigma_y$ are standard non-linearities such as *sigmoid* or *tanh*.

## 2.2 Graph Generation

A graph is given by $G = (\mathcal{V}, \mathcal{E})$ where $\mathcal{V}$ is a set of nodes (or vertices) and $\mathcal{E} \subseteq \mathcal{V} \times \mathcal{V}$ is a set of tuples denoting the nodes connected by an edge in the graph. Additionally, for each $v \in \mathcal{V}$, denote by $x_v \in \mathbb{R}^m$ the features of node $v$. Similarly, $e_{uv} \in \mathbb{R}^k$ denotes the features of the edge $(u, v) \in \mathcal{E}$. For example, in a molecular graph, nodes are atoms, and their features will contain the element; and edges correspond to bonds, and their features will contain the bond types (single, double, etc.).[2] Another example is of social networks, where nodes corresponds to users and their features to user profiles; and edges correspond to connections between users and their features contain metadata on this connection.

The topic of designing neural networks to operate specifically on graphs is dominated by Graph Neural Networks (GNNs) which mostly rely on a message-passing scheme to propagate information between nodes. While these architectures are extremely successful in node level and graph level prediction, they are not as prevalent in the context of graph generation, and many such approaches are restricted to small graphs (though (Davies et al., 2023) is a notable exception).

Formally, the task of graph generation is usually concerned with learning to model distributions: concretely, given a set of $N$ graphs $\{G_i\}_{i=1}^N$ originating from an underlying distribution $p$, the goal of graph generation is to devise an algorithm that generates new graphs from the underlying distribution $p$. Prior work has mostly adapted successful generative methods over a continuous space to the domain of graphs (Gómez-Bombarelli et al., 2018; Blaschke et al., 2018; Kadurin et al., 2017; Prykhodko et al., 2019). In this work we focus on using recurrent models which can be employed naturally to generate discrete objects. Crucially, Flam *et al.* (Flam-Shepherd et al., 2022) have shown that sequential generation is favorable compared to competing approaches in the context of molecular graph generation.

## 2.3 Sequential Graph Generation

When applying recurrent models for graph generation, the graph first needs to be "flattened" into a sequence. As there is no natural order for a graph, one must artificially induce such an order; for example, the approach taken by (You et al., 2018b) considers generation of breadth-first search (BFS) trajectories. While there are many ways to convert a graph into a sequence, in this work we focus on depth-first search (DFS); a strong motivation for this choice is that this is the method used to convert graph molecules into a linear representation called SMILES strings (Weininger, 1988). By convention, the output of the DFS algorithm is a spanning tree and we consider the induced order of the graph as the order in which the vertices were visited during the DFS run (also known as pre-order traversal).

In what follows we formally define the concepts discussed.

**Definition 2.1.** *Given a connected graph $G = (\mathcal{V}, \mathcal{E})$ with $|\mathcal{V}| = n$, we say the permutation $\pi \in \mathbb{S}_n$ **is a valid ordering of** $G$ if it is possible to run DFS over $G$ and visit the vertices in the order induced by $\pi$. Denote the sequence corresponding to a valid ordering $\pi$ of $G$ by*

$$\mathbf{s}(G, \pi) = (v_{\pi(1)}, \dots, v_{\pi(n)}). \tag{2.5}$$

*Denote the set of all such sequences for a given graph $G$ by*

$$\mathcal{S}(G) = \{\mathbf{s}(G, \pi) : \pi \text{ is a valid ordering of } G\}. \tag{2.6}$$

---

[1]Note that the bias term may be encapsulated into the input processing matrices by expanding the input with an additional dimension and assigning a fixed value of 1 on that coordinate.

[2]Molecular node and edge features may contain other properties as well.

Clearly, for a non-trivial graph $\mathcal{S}(G)$ will contain many sequences. In this work we have a special interest in sequences that share the same end vertex.

**Definition 2.2.** *Let $\mathcal{S}(G, v)$ denote all sequences terminating at node $v \in \mathcal{V}$, formally,*

$$\mathcal{S}(G, v) = \{\mathbf{s} \in \mathcal{S}(G) : s_n = v\} \tag{2.7}$$

*In case there are no valid DFS trajectories terminating as a certain node $v$, $\mathcal{S}(G, v) = \varnothing$ by definition.*

In the following section we discuss the desired properties for recurrent models when used for graph generation.

## 3 Structure Agnostic Recurrent Models

Recurrent models are a natural choice when generating discrete objects such as text. On the other hand, graphs are discrete objects with no naturally induced order. In Section 2 we described a mapping between graphs and sequences; and in particular, the fact that many different sequences correspond to the same graph. In this section we present our method that overcomes the issues described.

### 3.1 Generating Depth-First Search Traversals

In this work we use recurrent models to generate DFS traversals of graphs. Clearly when generating a DFS traversal, the next node to be generated depends on the nodes generated thus far and in particular the last generated node. An important observation is that the output of the recurrent model should be *invariant to different valid orderings corresponding to the same subgraph* as long as they lead to the same node. The following definition formalizes this notion,

**Definition 3.1.** *We say a recurrent model is **structure invariant** with respect to a connected graph $G$ if*

$$\forall v \in \mathcal{V}, \ \forall \mathbf{s}_1, \mathbf{s}_2 \in \mathcal{S}(G, v) \quad it\ is\ the\ case\ that \quad f_o(\mathbf{s}_1) = f_o(\mathbf{s}_2). \tag{3.1}$$

*If the above condition is satisfied for all $G \sim \mathcal{D}$, we say that the recurrent model is **structure invariant with respect to a distribution** $\mathcal{D}$.*

Figure 1 depicts a graph and two different DFS traversals sharing the same root and terminal node. A recurrent model processing the two DFS traversals will ideally generate the same node that will be attached to node $D$.

Definition 3.1 describes the structure invariance property with respect to a graph. Since recurrent models generate the traversal sequentially, we would like this property to hold at any moment during generation, i.e., we want to modify Definition 3.1 to take into account partial DFS traversals.

**Definition 3.2.** *For a connected graph $G$, we say a connected subgraph $\tilde{G} \subseteq G$ is **induced by DFS over** $G$ if there exists a valid ordering $\pi \in S_n$ of $G$, and $k \leq n$ such that $(v_{\pi(1)}, \ldots, v_{\pi(k)})$ is a valid ordering of $\tilde{G}$. Denote the set of all DFS induced subgraphs over $G$ by $\mathcal{G}_{DFS}(G)$.*

At this stage, a reader might question the necessity of Definition 3.2 and why $\mathcal{G}_{DFS}(G)$ differs from the set of all connected subgraphs of $G$. We note that for a general connected graph $\mathcal{G}_{DFS}(G)$ does **not** correspond to the set of all connected subgraphs.

**Proposition 3.3.** *For a connected graph $G$,*

$$\mathcal{G}_{DFS}(G) \neq \left\{ \tilde{G} \mid \tilde{G} \subseteq G \ and \ \tilde{G} \ is \ connected \right\} \tag{3.2}$$

Figure 2 depicts a graph and two connected subgraphs, one which is induced by DFS and the other that cannot be obtained by a DFS traversal.

With the notion of DFS induced subgraphs in hand, we now present the following definition of *total structure invariance*:

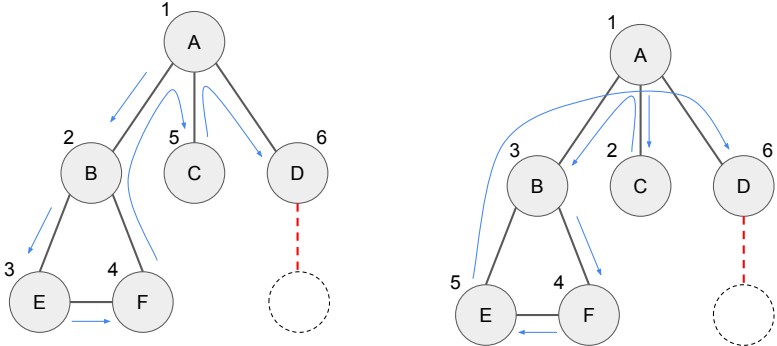

Figure 1: Illustration of two DFS traversals of the same graph starting from node $A$ and terminating at node $D$, blue lines denote traversal order. (Left) traversal resulting in the sequence $A(BEF)(C)D$. (Right) traversal resulting in the sequence $A(C)(BFE)D$. The parentheses denote the opening and closing of branches when traversing the tree; with this syntax it is possible to reconstruct the tree from such sequences. Note that multiple sequences correspond to the same tree, a fact that lies at the heart of this work.

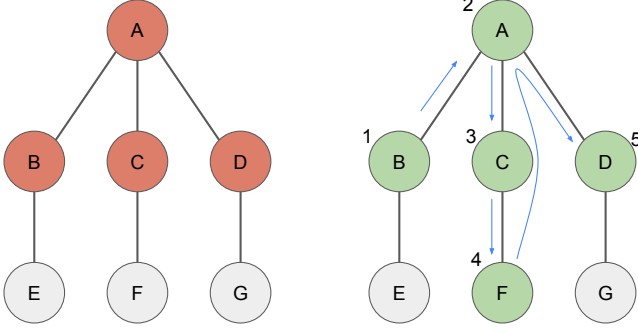

Figure 2: Illustration of the same graph with two connected subgraphs: (Left) subgraph which is not induced by DFS. (Right) subgraph induced by DFS, arrows depict a traversal resulting in the sequence $BA(CF)D$.

**Definition 3.4.** *We say a recurrent model is **totally structure invariant** with respect to a connected graph $G$, if*

$$\forall \tilde{G} \in \mathcal{G}_{DFS}(G), \ \forall v \in \mathcal{V}(\tilde{G}), \ \forall \mathbf{s}_1, \mathbf{s}_2 \in \mathcal{S}(\tilde{G}, v) \ \text{ it is the case that } \ f_o(\mathbf{s}_1) = f_o(\mathbf{s}_2). \quad (3.3)$$

*If the above condition is satisfied for all $G \sim \mathcal{D}$, we say that the recurrent model is **totally structure invariant with respect to a distribution $\mathcal{D}$**.*

Note that per Definition 2.2, the condition vacuously holds for non terminal nodes.

In the next section we discuss how to train recurrent models which are totally structure invariant with respect to a given training distribution over graphs.

## 3.2 Regularizing Towards Total Structure Invariance

Motivated by the observation discussed in Section 3.1, we propose training recurrent models that are totally structure invariant with respect to the underlying distribution over graphs. It would be appealing to characterize the class of all totally structure invariant functions and optimize over those. Unfortunately, it is difficult to attain a crisp characterization of structure invariance as this property depends on the training distribution.

Instead, we propose encouraging total structure invariance via regularization. Specifically, we would like to minimize the following auxiliary loss,

$$\mathbb{E}_{G \sim \mathcal{D}} \mathbb{E}_{\tilde{G} \sim \mathcal{G}_{DFS}(G)} \mathbb{E}_{v \in \mathcal{V}(\tilde{G})} \mathbb{E}_{\mathbf{s}_1, \mathbf{s}_2 \in \mathcal{S}(\tilde{G}, v)} \left[ (f_o(\mathbf{s}_1) - f_o(\mathbf{s}_2))^2 \right] \tag{3.4}$$

which we refer to as *Orderless Regularization* (OLR). Examining Equation 3.4, we note that sampling from $\mathcal{G}_{DFS}(G)$ is easily done by randomly selecting a root node and running DFS with stochastic decision making. On the other hand, given $\tilde{G}$ and $v$, sampling from $\mathcal{S}(\tilde{G}, v)$ is hard and has been shown to be NP-complete (Beisegel et al., 2019).

## 3.3 Sampling Trajectories with Common End Vertex

The problem of generating all DFS trajectories that terminate at the same vertex is hard and there are no known efficient algorithms for this task. In order to overcome this obstacle we apply a heuristic for computing such trajectories. We highlight that our proposed scheme is not equivalent to a uniform sampling over all possible trajectories; however, in Section 5 we show that the resulting regularization scheme is effective empirically.

Next, we formally show that for practical graphs there exists efficient algorithms to generate such trajectories.

**Definition 3.5.** *Let $G = (\mathcal{V}, \mathcal{E})$ be an arbitrary graph. $G$ is said to be $k$-**edge-connected** if the subgraph $G' = (\mathcal{V}, \mathcal{E} \backslash \tilde{\mathcal{E}})$ is connected for all $\tilde{\mathcal{E}} \subseteq \mathcal{E}$ such that $|\tilde{\mathcal{E}}| < k$ and $\exists \tilde{\mathcal{E}}$ s.t. $|\tilde{\mathcal{E}}| = k$ and $G'$ is not connected.*

**Proposition 3.6.** *There is an efficient algorithm to find distinct DFS trajectories with common end vertex for any $k$-edge connected graph for $k \leqslant 2$.*

We note that in many real world tasks, graph are rarely $k$-edge connected for $k > 2$. For example, in the ZINC molecular dataset, more than 99.5% of molecular graphs are 1-edge connected.

*Proof Sketch.* Find a min-cut: by the definition of 1-edge-connected graphs the min-cut includes a single crossing edge $(u, v)$. By removing $(u, v)$ the graph is partitioned into two connected components, $G_1$ and $G_2$ containing $u$ and $v$ respectively. Run a DFS on $G_1$ with $u$ as the root vertex to result in $(u_1, \ldots, u_k)$, and similarly for $G_2$ to result in $(v_1, \ldots, v_m)$ (where $v_1 = v$ and $u_1 = u$).[3] We can now construct a DFS traversal on $G$ by 'gluing' together the sequences as,

$$(v_1, u_1, \ldots, u_k, v_2, \ldots, v_m) \tag{3.5}$$

We can run another (stochastic) DFS on $G_1$ from $u$ to obtain $(u_{\tilde{\pi}(1)}, \ldots, u_{\tilde{\pi}(k)})$ where $\pi \in S_k$ and $\tilde{\pi}(1) = 1$. We can construct a second DFS sequence as in Equation 3.5,

$$(v_1, u_{\tilde{\pi}(1)}, \ldots, u_{\tilde{\pi}(k)}, v_2, \ldots, v_m) \tag{3.6}$$

We have created two valid DFS sequences that both terminate at $v_m$. □

---

[3] $k$ and $m$ denote the size of partitions and satisfy $k + m = |\mathcal{V}|$.

See Appendix A for the full details and the case of 2-edge connected graphs. Note that our method for generating distinct DFS trajectories is not exhaustive and there may be additional trajectories not detected via the algorithm induced by the proof sketch.

## 4    Related Work

In this section we discuss several relevant topics to graph generation. For a comprehensive review on graph generation see (Guo & Zhao, 2022; Zhu et al., 2022).

**One Shot Generation**    Classic generative architectures (e.g. Variational autoencoders (VAEs) (Kingma & Welling, 2013), Generative adversarial networks (GANs) (Goodfellow et al., 2020), etc.) work by learning a continuous mapping from a latent distribution to generate new examples with similar properties to the training distribution. These models usually incorporate a neural architecture that maps directly from the latent space to the domain of the training data (e.g. images) and therefore the output space must be predetermined. These properties pose a challenge when applied to the domain of graphs, as the latter are discrete objects with variable size and no naturally induced order. In order to circumvent these caveats, prior work (Assouel et al., 2018; De Cao & Kipf, 2018; Du et al., 2022a; Fan & Huang, 2019; Flam-Shepherd et al., 2020; Guo et al., 2020; Honda et al., 2019; Ma et al., 2018; Madhawa et al., 2019; Shi et al., 2020; Simonovsky & Komodakis, 2018; Zou & Lerman, 2019) has used a one-shot generation strategy. That is, the output space is limited by design to a specific representation of graphs (i.e. adjacency matrix or adjacency list) of specific size and the output is generated in a single forward pass. While the one-shot strategy has its merits, there are a few significant drawbacks such as the inability to generate graphs with arbitrary large number of nodes.

**Sequential Generation**    The idea of using autoregressive models for graph generation is not new and there have been several works in this vein. GraphRNN (You et al., 2018b) proposes generating BFS trajectories in order to limit the number of possible orderings per graph. Other works take a different approach of generating edges in an autoregressive manner (Bacciu et al., 2020; Goyal et al., 2020). Additional approaches include MolecularRNN (Popova et al., 2019) which incorporates a reinforcement learning environment to generate nodes and edges sequentially. Yet another approach includes sequentially generating subgraph structures (Jin et al., 2018; Liao et al., 2019; Podda et al., 2020). Another recent work (Bu et al., 2023) treats the induced order as a problem of dimensionality reduction and attempts to learn mappings from graphs to sequences. In this work we argue that the most effective inductive bias for the use of autoregressive models to generate graphs is to be invariant to different orderings possible under the training distribution.

**Molecule Generation**    One of the most prominent uses of graph generation, which is used for evaluation in this work, is that of molecule generation. Molecular generation is applicable to the development of synthetic materials, drug development and more. Molecules are 3D objects which are naturally represented as point clouds[4] with corresponding geometric approaches (Garcia Satorras et al., 2021; Simm et al., 2020a;b; Hoogeboom et al., 2022) which utilize inherent symmetries in the architectures employed. While 3D representations are richer and carry significant information that does not transfer to 1D and 2D representations, they are costly to obtain and therefore the corresponding amount of data is limited as compared to 1D and 2D representations, which are ubiquitous. Another aspect of molecule generation is when the generation is conditioned to satisfy certain properties. For example, (Skalic et al., 2019; Zhang et al., 2022; Rozenberg & Freedman, 2023) generate molecules that are conditioned to bind to specific ligand structures, (Kang & Cho, 2018; Zang & Wang, 2020) generate molecules that fulfill certain chemical properties. In this work we consider the task of de-novo generation (Arús-Pous et al., 2019; Lim et al., 2018; Pogány et al., 2018; Tong et al., 2021) where the objective is to generate molecules with similar properties to those in the training data.

**Permutation Invariant Recurrent Models**    Another relevant topic is the use of autoregressive models for problems over sets which, like graphs, lack a natural order. There have been many works focusing on

---

[4]In a point-cloud representation of a molecule each point represents an atom and bonds are implicit from the distances between atoms.

Table 1: Wiener Index results for training with and without OLR. We report Mean Absolute Error (MAE) and accuracy (computed by rounding the output to the nearest integer). For both metrics considered, training with OLR dramatically improves performance.

|  | MAE ($\downarrow$) | Accuracy ($\uparrow$) |
|---|---|---|
| Vanilla | 2.24 (0.12) | 0.18 (0.02) |
| OLR | **1.32 (0.10)** | **0.28 (0.04)** |

problems over sets. The most prominent of these is DeepSets (Zaheer et al., 2017) which applies a deep neural network on each element of the set and then aggregates the result with a permutation invariant operator (e.g. sum or max), finally applying another deep neural network on the aggregated result. There have also been autoregressive works designed for sets: Murphy et al. (2018) use RNNs on different permutations and output the average. While this requires $n!$ orderings for a set of size $n$, the authors have presented several approximation techniques and justified them empirically. Cohen-Karlik et al. (2020) have shown that while DeepSets are universal, some permutation invariant functions require unbounded width to implement successfully and have proposed using RNNs with a regularization term which enforces permutation invariance. In this work we extend the concepts introduced in previous works into the realm of drug design and sequential graph generation, where a desired property of models is to hold invariance for certain permutations as induced by the data distribution. In the work of Cohen-Karlik et al. (2020) the regularization is geared towards fully permutation invariant models; that is, their work may be viewed as a specific case of graph-aware regularization where the data is represented by fully connected graphs. In this work we generalize these concepts and formalize the problem for graphs with more general structures. As a result, the straightforward regularization term proposed in (Cohen-Karlik et al., 2020) cannot be used, and a more sophisticated regularization scheme is required. Specifically, using the lens of DFS trajectories of graph, we suggest regularizing over valid sequences and devise an efficient approximation for generating such sequences.

## 5 Experiments

### 5.1 Wiener Index

In order to gauge the effectiveness of OLR, we conducted a straightforward experiment designed to predict the Wiener index of graphs. The Wiener index is a topological metric for molecules, involving the summation of distances between all pairs of vertices within a given graph. These graphs are symbolically represented as strings, using parentheses as exemplified in Figure 1.[5] We employed a Long Short-Term Memory (LSTM) model with a hidden width of 100, trained as a regression task. During training, we used graphs containing 10 nodes and a training set consisting of 50 examples; our aim was to determine the effectiveness of OLR in the case when data is extremely scarce. The network was trained until convergence with perfect training accuracy and evaluated on a test set consisting of 200 data points. We report the average mean absolute error and accuracy as computed by rounding the networks output. As can be seen in Table 1, using OLR improves results significantly.

To further investigate the effect of training with OLR, we visualize the embeddings of the hidden state when the inputs are different representations drawn from two different graphs. A desired property is that different representations of the same graph are clustered together; we show that this phenomenon does indeed occur, as illustrated in Figure 3. This experiment demonstrates that the training with OLR results in models that are significantly more invariant to different orderings of the same graph, a desired property for models trained for the task of graph generation.

---

[5]As the Wiener index of a graph does not involve node features, we omit the node labeling from the representation which yields a string of only opening and closing parentheses.

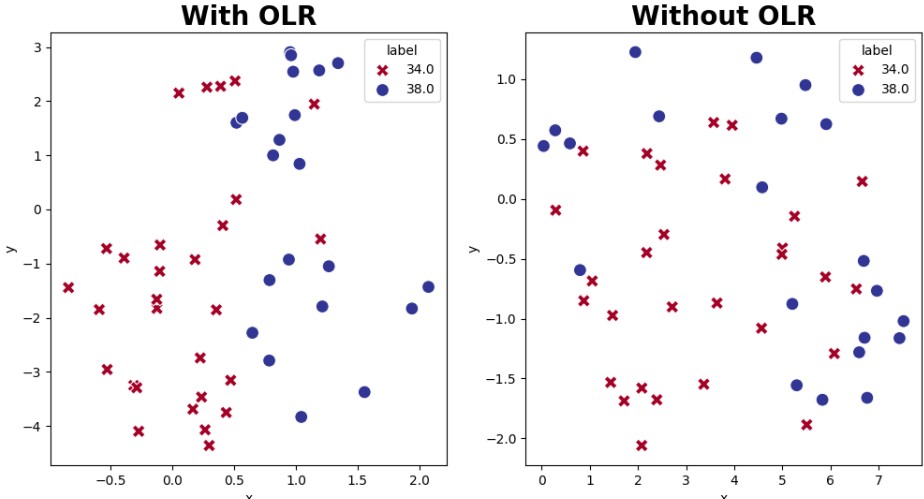

Figure 3: t-SNE visualizations of hidden states when training with and without OLR. Data is generated as different representations of two graphs (i.e. different DFS trajectories of the same graph), the first with a Wiener index of 34, and the second with 38. (Left) training with OLR yields hidden states that are clearly clustered into two groups. (Right) training without regularization, there is no apparent separation in the embedding space between the two groups.

## 5.2 De Novo Molecule Generation

A prominent application of graph generation is that of molecule design. Graph generation tasks range from de novo generation where the objective is to generate molecules with similar properties to a given dataset, to conditional generation for which the task is to generate a graph given a second graph with specific characteristics, i.e. a ligand that binds to a specific target. Our empirical evaluation focuses on the former. We evaluate our proposed regularization method on the MOSES benchmark (Polykovskiy et al., 2020) and compare to relevant baselines. Our implementation is based on the work of CharRNN which use three layers of the LSTM architecture each with hidden dimension of 600 (for complete details refer to (Segler et al., 2018)). We find a consistent improvement when adding OLR to the objective of autoregressive models.

The data curated by (Polykovskiy et al., 2020) is refined from the ZINC dataset (Sterling & Irwin, 2015) which contains approximately $4.6M$ molecules. The authors filter the data based on molecular weights, number of rotational bonds, lipophilicity, etc. to result in a total of $2M$ molecules. The authors provide partitions of the data into train, test and scaffold test to allow fair evaluation.[6]

**Computing Trajectories** OLR works by feeding two different trajectories that terminate at the same node. While this calculation is feasible to perform during the forward pass it introduces a computational bottleneck. In order to circumvent this issue we employ the following calculations offline. For each molecule we first index all min-cuts and randomly select one. We then generate multiple (10) traversals terminating at the same node as described in Section 3.3 and write the sequences into a file along with the original molecule from which the trajectories are derived from. When loading the data, two trajectories are selected at random and used as inputs to the OLR objective described in Section 3.2.

**Data Filtering** Our offline computation of trajectories in Section 5.2 requires that there are min-cuts that induce sufficient number of different DFS traversals terminating at the same node. While 99.9% of the molecules in MOSES have at least two such trajectories, we filter the data to remain with molecules

---

[6]The scaffold of a molecule is the structure induced by its ring systems along with the connectivity pattern between these systems. The scaffold test partition contains molecules with structures that did not appear in the train and test partitions. The scaffold test allows for the evaluation of how well the model can generate previously unobserved scaffolds.

Table 2: Generation results at validity threshold of 0.8 for LSTM and GRU architectures. Leading result highlighted in bold for each metric. Rank Average is the average position of each method over all metrics considered. As can be seen, OLR outperforms the baselines considered for both architectures. Refer to the text for further details.

| Metric | LSTM | | | GRU | | |
|---|---|---|---|---|---|---|
| | Canonical | Rand. | OLR + Rand. | Canonical | Rand. | OLR + Rand. |
| Unique@1K (↑) | 0.8930 | **1.0** | **1.0** | 0.805 | **1.0** | **1.0** |
| Unique@10K (↑) | 0.6182 | 0.9975 | **0.9981** | 0.5314 | 0.9965 | **0.9967** |
| FCD/Test (↓) | 1.1208 | 0.8568 | **0.7784** | 1.2616 | **0.9602** | 0.9752 |
| SNN/Test (↑) | **0.5599** | 0.4967 | 0.4936 | **0.5718** | 0.4948 | 0.4978 |
| Frag/Test (↑) | 0.9953 | 0.9947 | **0.9958** | 0.9945 | 0.9940 | **0.9961** |
| Scaf/Test (↑) | 0.6370 | **0.8246** | 0.8220 | 0.5857 | **0.8252** | 0.8159 |
| FCD/TestSF (↓) | 1.8318 | 1.4236 | **1.3089** | 1.921 | 1.7269 | **1.7022** |
| SNN/TestSF (↑) | **0.5234** | 0.4795 | 0.4769 | **0.5352** | 0.4755 | 0.4785 |
| Frag/TestSF (↑) | 0.9920 | 0.9919 | **0.9926** | 0.9911 | 0.9908 | **0.9932** |
| Scaf/TestSF (↑) | 0.0245 | **0.1185** | 0.0931 | 0.0250 | 0.1028 | **0.1123** |
| IntDiv (↑) | 0.8527 | 0.8508 | **0.8537** | 0.8499 | **0.8535** | 0.8531 |
| IntDiv2 (↑) | 0.8457 | 0.8449 | **0.8479** | 0.8424 | **0.8475** | 0.8471 |
| Filters (↑) | **0.9889** | 0.9705 | 0.9702 | **0.9908** | 0.9678 | 0.9702 |
| Novelty (↑) | 0.8969 | 0.9797 | **0.9809** | 0.8787 | **0.9787** | 0.9748 |
| **Rank Average** | 2.28 | 2.07 | **1.57** | 2.42 | 1.92 | **1.57** |

which have at least 10 different trajectories satisfying the criteria defined. After filtering we are left with approximately 500K molecules for training, and 55K for test and scaffold test partitions. We note that in following sections we show our method is most effective when training data is scarce and therefore the filtering process does not limit the applicability of our proposed regularization scheme.

**Results** Our results for training with OLR compared to other baselines trained on the same data are shown in Table 2. The most relevant baselines is CharRNN (Segler et al., 2018) which is an autoregressive model trained on Canonical SMILES. We further compare to a randomized version of CharRNN inspired by the finding of (Arús-Pous et al., 2019) which show that augmenting the data by using randomly generated SMILES representations of the same molecule improves performance. We also attempted to compare our method to other non-autoregressive models such as those based on Variational Autoencoders (VAEs) (Blaschke et al., 2018; Gómez-Bombarelli et al., 2018; Kadurin et al., 2017); however, we found that the models did not produce valid molecules when trained with 1000 examples, so we do not report these results. We use the metrics defined by the MOSES benchmark (Polykovskiy et al., 2020); see Appendix B for a thorough description of these metrics.

In order to demonstrate the effectiveness of OLR we use 1000 randomly sampled data points from the training set and evaluate over the entire test set. When training with small amounts of data there is a trade-off between the validity of the generated molecules and the uniqueness and other metrics. Our evaluation considers the best performing models for each method providing the validity of the generated molecules exceeds 80%.

Results are depicted in Table 2. As can be seen, adding randomized variants of the molecules outperforms the original work of (Segler et al., 2018) which train an RNN as a language model using only canonical SMILES. Furthermore, adding the OLR objective exceeds the performance of randomized SMILES. In order to clearly depict the performance difference, we calculate the rank of each method on each metric considered. The average rank of each method is added as the last row of Table 2.

# 6  Discussion

Graph generation poses unique challenges due to the discrete and unordered nature of graphs, which differ from continuous data typically handled by generative models. While various generative approaches such as GANs, VAEs, and autoregressive models have been successful in other domains, their application to graph generation requires careful consideration of the inherent structural complexities. Our work focuses on sequential graph generation using autoregressive architectures, motivated particularly by applications in molecular graph generation.

The choice of depth-first search (DFS) trajectories as the representation for graph sequences offers a structured approach aligned with the nature of graph exploration and has relevance in chemical informatics where molecules are often represented as SMILES sequences. However, the multitude of possible DFS trajectories for a given graph poses a challenge when devising a regularization scheme to ensure model invariance.

The introduction of Orderless Regularization (OLR) addresses this challenge by promoting model robustness against different DFS orderings of the same sub-graph. By incorporating OLR into the training process, we mitigate the dependency on specific DFS trajectories and enhance the generalization capabilities of the autoregressive model. This regularization term proves particularly beneficial in scenarios with limited training data, as demonstrated empirically in our study on small molecule generation.

The computational aspect of generating DFS trajectories with a common end-vertex, a prerequisite for training with OLR, presents a notable challenge. However, our devised algorithm efficiently tackles this challenge under specified constraints, facilitating effective training with OLR.

While our approach shows promise in small molecule generation, its generalizability to larger and more complex graphs warrants further investigation. Scalability issues may arise with increasingly heterogeneous graphs, necessitating advancements in algorithmic techniques or adaptations of OLR. Additionally, the dependence on specific constraints for generating DFS trajectories highlights a potential limitation, prompting exploration of alternative regularization techniques or extensions of OLR to handle diverse graph structures.

An avenue for future research involves investigating the suitability of OLR for diverse autoregressive architectures, including Transformers. Adapting OLR to Transformer-based models necessitates adjustments due to the absence of a hidden state, which serves as the foundation for enforcing invariance in recurrent architectures. Furthermore, Transformers are permutation invariant architectures by design. A straightforward approach to encorporate OLR in Transformers would be to add positional encodings to the sequences and enforce invariance by penalizing the gap in the output of the model over different sequences representing the same graph. To test the applicability of OLR for Transformers, we repeat the Wiener index experiment (Section 5.1) with the recurrent architecture replaced by a Transformer. We find that regularization does not yield an improvement in results for an architecture with a comparable number of parameters: the accuracy is 0.12 compared to 0.28 achieved by an LSTM with OLR. One explanation for the gap in performance is the fact that regularization is performed on partial sequences which admit only certain positional embeddings and the model cannot extrapolate the desired behaviour to positional embeddings corresponding to unseen string orders.

These results indicates that Transformers require a different approach to adequately regularize towards structure invariance. A possible method for biasing Transformers towards graph invariance would be to design positional encodings that are aware of the structure of the graph, an approach which may have connections to Attention GNNs (Velivckovic et al., 2017; Zhang et al., 2018; Lee et al., 2018). Consequently, the extent to which Transformer models can leverage the advantages of OLR or other regularization methods remains uncertain and is a promising direction for future work.

# 7  Conclusions

In this work we highlight the innate gap that every autoregressive model for graph generation must mitigate - the induced order on graphs. We propose a different approach to previous works by introducing a novel regularization scheme that encourages learning hypotheses that are invariant to different DFS orderings. We demonstrate empirically that our proposed method improves performance for autoregressive models and is

especially effective when the available datasets are small, as is the case in many real world problems. We believe that our approach can contribute to the applicability of autoregressive models such (e.g. State-space models) for graph generation and that similar ideas may be incorporated in various generation strategies beyond the scope of this work.

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

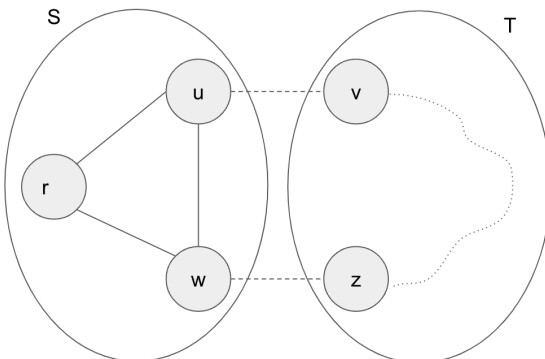

Figure 4: Proof illustration - $S$ has a cycle and two different trajectories starting from $u$ and ending with $w$ ($urw$ and $uw(r)$). Concatenating with the trajectory from $z$ to $v$ we obtain two different DFS trajectories with a shared suffix.

Chengxi Zang and Fei Wang. Moflow: an invertible flow model for generating molecular graphs. In *Proceedings of the 26th ACM SIGKDD international conference on knowledge discovery & data mining*, pp. 617–626, 2020.

Jiani Zhang, Xingjian Shi, Junyuan Xie, Hao Ma, Irwin King, and Dit-Yan Yeung. Gaan: Gated attention networks for learning on large and spatiotemporal graphs. *arXiv preprint arXiv:1803.07294*, 2018.

Zaixi Zhang, Yaosen Min, Shuxin Zheng, and Qi Liu. Molecule generation for target protein binding with structural motifs. In *The Eleventh International Conference on Learning Representations*, 2022.

Yanqiao Zhu, Yuanqi Du, Yinkai Wang, Yichen Xu, Jieyu Zhang, Qiang Liu, and Shu Wu. A survey on deep graph generation: Methods and applications. *arXiv preprint arXiv:2203.06714*, 2022.

Dongmian Zou and Gilad Lerman. Encoding robust representation for graph generation. In *2019 International Joint Conference on Neural Networks (IJCNN)*, pp. 1–9. IEEE, 2019.

## A  Missing Proofs

In this section we show how to construct distinct DFS trajectories with common end vertex for a 2−edge connected graph conditioned that the graph is not a cycle.

*Proof.* From our assumption that the graph is not a cycle, there exists at least two nodes with degree $\geqslant 3$. Denote by $C = (S, T)$ a minimal cut of size 2 (such a cut exists from our assumption that the graph is 2-connected). Denote the edges of the minimal cut by $e_1 = (u, v)$ and $e_2 = (w, z)$ such that $u, w \in S$ and $v, z \in T$. Next, we claim that at least one of the partitions contains a cycle, otherwise there is a path connecting $S$ and $T$ since there are nodes in the graph which have a degree of 3 in the original graph with a path between them. Assume with out loss of generality that $S$ is the partition with a cycle, therefore there are at least 2 different traversals of $S$ that start with $u$ and end with $w$. There is also a trajectory between $z$ and $v$. Putting together, there are at least 2 trajectories of the entire graph with a common suffix which is the traversal of $T$. Figure 4 illustrates the proof concept. $\square$

## B  Metric Details

In this section we provide details for the metrics reported in Table 2.

A few of the similarity measures (SNN and IntDiv) are based on the *Tanimoto coefficient.* In order to compute the Tanimoto coefficient, the molecules are mapped to a vector of fingerprints where each bit in the vector represents the presence (or absence) of a specific fragment.[7] For molecules $A, B$, denote their fingerprints by $m_A$ and $m_B$ respectively, the Tanimoto coefficient is then calculated as the Jaccard index of the two vectors,

$$J(m_A, m_B) = \frac{|m_A \cap m_B|}{|m_A \cup m_B|} = \frac{|m_A \cap m_B|}{|m_A| + |m_B| - |m_A \cap m_B|}. \tag{B.1}$$

We denote the Tanimoto coefficient of molecules $A, B$ by $T(A, B)$.

**Unique@K** report the fraction of uniquely generated valid SMILES strings amongst the $K$ molecules generated (validity is determined by the RDKit library). We generate $30,000$ molecules for each model and report for $K = 1,000$ and $K = 10,000$. High uniqueness values ensure the models do not collapse into repeatedly producing the same set of molecules.

**Fréchet ChemNet Distance (FCD)** is a metric for evaluating generative models in the chemical context, the method is based on the well established *Fréchet Inception Distance* (FID) metric used to evaluate the performance of generative models in computer vision (Heusel et al., 2017).

Fréchet distance measure the Wasserstein-2 distance (Vaserstein, 1969) from the distributions induced by taking the activations of the last layer of a relevant deep neural net. In the case of FCD, molecule activations are probed from ChemNet (Mayr et al., 2018). Given a set of generated molecules, denote by $G$ the set of vectors as obtained by the activations of ChemNet, one can calculate the mean and covariance $\mu_G$ and $\Sigma_G$. Similarly, denote $\mu_R$ and $\Sigma_R$ the mean and covariance of the set of molecules in the reference set, the FCD is calculated as follows,

$$FCD(G, R) = \|\mu_G - \mu_R\|^2 + Tr\left(\Sigma_G + \Sigma_R - 2(\Sigma_G \Sigma_R)^{1/2}\right). \tag{B.2}$$

where $Tr(M)$ denotes the trace of the matrix $M$. Low FCD values indicate that the generated molecules distribute similarly to the reference set.

**Similarity to Nearest Neighbor (SNN)** is the average of the Tanimoto coefficient of the generated molecule set denoted by $G$ and their respective nearest neighbor in a reference set of molecules denote by $R$. High SNN indicates the generated molecules have similar structures to those in the reference set. This metric is in the range of $[0, 1]$.

**Fragment similarity (Frag)** is a fragment similarity measure based on the BRICS fragments (Degen et al., 2008). Denote the set of BRICS fingerprints vectors of the generated molecules by $G$ and similarly $R$ for the reference molecules. The fragment similarity is defined as the cosine similarity of the sum vectors,

$$Frag(G, R) = cosine\left(\sum_{g \in G} g, \sum_{r \in R} r\right) \tag{B.3}$$

The Frag measure is in the range of $[0, 1]$, values closer to 1 indicate that the generated and reference molecule set have a similar distribution of BRICS fragment.

**Scaffold similarity (Scaff)** is similar to the fragment similarity, instead of the BRICS fragment, Scaff is based on mapping molecules to their Bemis–Murcko scaffolds (Bemis & Murcko, 1996).[8] The measure also has a range of $[0, 1]$, values closer to 1 indicate that the generated molecule set has a similar distribution of scaffold to the reference set.

---

[7]The molecular fingerprints are obtained from RDKit (Landrum, 2006) and are based on the extended-connectivity fingerprints (Rogers & Hahn, 2010).

[8]Bemis–Murcko scaffold is the ring structure of a molecule along with the bonds connecting the rings, i.e. the molecule without the side chains.

**Internal diversity (IntDiv)**   is a mesure of the chemical diversity within a generated set of molecules $G$. This metric indicates

$$IntDiv_p = 1 - \left( \frac{1}{|G|^2} \sum_{A,B \in G} T(A,B)^p \right)^{1/p} \tag{B.4}$$

We report the internal diversity for $p = 1, 2$. This measure is in the range $[0, 1]$. Low values indicate a lack of diversity in the generated molecules, i.e. that the model outputs molecules with similar fingerprints.

**Filters**   is the fraction of generated molecules that pass a certain filtering that has been applied to the training data. The metric is in the range of $[0, 1]$, high values indicate that the model has learnt to generate molecules which avoid the structures omitted by the filtering process.

**Novelty**   is the fraction of generated molecules that does not appear in the training set. This measure is in the range of $[0, 1]$ and is an indication of the whether the model overfits the training data.

