# OpenReview forum: "Overcoming Order in Autoregressive Graph Generation for Molecule Generation"
_TMLR — Accepted by TMLR_

### Review · Reviewer_kS9K · 2024-03-04

**Summary Of Contributions:**

This paper proposes a new regularization scheme for sequential graph generation by recurrent neural networks, called Orderless Regularization (OLR), that encourages the hidden state of the RNN to be invariant to different node orderings in the training graphs. Experiments show OLR improves the performance sequential graph generation models.

**Audience:**

Yes

**Claims And Evidence:**

Yes

**Requested Changes:**

*A key question that I have is how do we know the representations we obtain with OLR regularization is invariant? The performance improvements in the regression and graph generation tasks could just be due to regularization, and not due to any invariance imbued in the representations. For example, if one were to encode pairs of molecule SMILES that encode the same graph structure but are from different orderings, would we expect a more similar pair of embeddings with OLR?

*I would like some more clarity on what the input graphs are for each of the evaluation settings. The molecule generation task makes more sense to me (SMILES of a set of molecules curated by Polykovskiy). What about the Wiener index task? How are these graphs generated? And are the graphs represented simply as text codes?

*In the Wiener index task, the dataset size is restricted to be very small (50 training, 200 test). How does the performance gap of vanilla and vanilla + OLR change as the dataset sized is increased, eg to 500, 5000, 50000 training examples. (I just arbitrarily picked some numbers)

*Table 2: Would be good to see Canonical + OLR to see the effect of the OLR regularization in the absence of any input data augmentation.

**Strengths And Weaknesses:**

Strengths:

*Paper tackles an interesting problem for sequential graph generation

Weaknesses:

*Experimental setup is a bit unclear

*Some concerns about the evaluation

---

> ### Author Response · Authors · 2024-04-24
> **Author response**
>
> Thank you for the comments and points raised. We address each concern below.
> 1. In order to show that training with OLR indeed yields models that are invariant to different representations of the same graph we added a t-SNE visualization of the hidden states in the Wiener experiment.  In particular, we examine the hidden states of different sequences representing the same graph, and compare those with hidden states of sequences representing a different graph. The plots depict that when training without regularization, the hidden states of all sequences - whether they represent the same graph or not - are scattered in a uniform fashion.  By contrast, when training with OLR the visualization clearly shows a structure in the embedding space: hidden states of sequences corresponding to the same graph are clustered together, and separated from those representing different graphs.  This is described in Section 5.1.
> 2. Indeed the molecular experiments use the SMILES strings as inputs which are later tokenized according to the MOSES benchmark implementation. For the Wiener index experiment, the inputs are DFS trajectories of general graphs (with no specific order over the nodes). As every DFS trajectory can be encoded as described in Figure 1, we simply omit the node names and reside with a sequence of opening and closing parenthesis. In order to generate the data we make use of the random_tree function provided by the networkx python library. We have added a footnote to clarify this point.
> 3. The benefit of the regularization method is evident when the amount of data available for training is small, when increasing the size of the training data the performance gap shrinks.  We refer to the advantages presented by OLR in the scarce data regime in the abstract as well in Sections 5.1 and 5.2.
> 4. The problem with performing a Canonical + OLR sub-experiment is that this would contradict the fundamental logic behind the canonical representation.  Recall that the canonical representation enforces a *unique* representation for each SMILES string; and consequently assumes that a model should implicitly learn the canonicalization mechanism.  OLR is only relevant for scenarios in which there is no prescribed order for the graph flattening mechanism.

---

### Review · Reviewer_mLu8 · 2024-03-04

**Summary Of Contributions:**

The paper titled "Overcoming Order in Autoregressive Graph Generation" presents a new approach to graph generation that addresses the challenge of order invariance in sequential graph generation models. It introduces an Orderless Regularization (OLR) term to recurrent neural networks (RNNs) that encourages the model to learn representations invariant to different orderings of graph sequences. This is particularly useful in applications like molecular graph generation where the sequence order can be arbitrary. The authors demonstrate the effectiveness of OLR through experiments on the Wiener index prediction and de novo molecule generation, showing that it improves performance, especially in data-scarce scenarios.

**Audience:**

Yes

**Claims And Evidence:**

Yes

**Requested Changes:**

1. It's crucial to broaden the experimental validation by comparing the OLR method with more baseline models, incorporating different types of RNNs, and utilizing a variety of datasets. This would provide a more comprehensive understanding of the method's performance across different contexts and improve confidence in its generalizability and effectiveness. To deepen the experimental validation of the OLR method, the authors should aim to answer the following questions, either through theoretical analysis or additional experiments:
- How does the OLR method compare with state-of-the-art models across different domains? This would involve not just other RNN-based approaches but also models that might use different underlying architectures or methodologies for graph generation.
- What is the impact of using different RNN architectures (such as LSTM, GRU, etc.) on the effectiveness of the OLR method?
- Can the OLR method maintain its performance across diverse datasets, including those with varying sizes, complexities, and domain-specific characteristics? This could involve testing on datasets from different fields, such as social network graphs, biochemical molecule structures, and communication networks, to evaluate the method's robustness and flexibility.
- What are the computational requirements and scalability of the OLR method across different settings? Evaluating the method's performance in terms of time and resource efficiency, especially as the size of the graph increases, is essential for assessing its practical applicability.

2. Add a detailed comparison with the regularization approach introduced by  Cohen-Karlik et al., 2020. Highlighting the specific advancements or differences brought by OLR is essential to clarify its novelty and contribution to the field.

3. The paper would greatly benefit from including a discussion about the potential integration and comparison of the OLR method with Transformer models. This should explore how the OLR approach might be adapted for or compared with Transformer architectures, which have shown considerable success in sequence and relational data modeling.

**Strengths And Weaknesses:**

## Strengths

1. Addressing the order invariance issue in graph generation is a critical challenge, as the sequence in which nodes are generated can significantly impact the performance of generative models. The paper introduces a simple solution by adding an auxiliary loss term, to mitigate the dependence on node sequence order.
2. The paper is well-written, offering clear and concise explanations of the motivation and the proposed methodology.

## Weaknesses

1.  The limited experimental evaluation in the paper is a significant shortcoming. The comparison is made against just one baseline model, which does not fully test the strengths and potential weaknesses of the OLR approach against the array of methods available in the field. Moreover, the choice to use only one type of RNN and to conduct experiments on a single dataset restricts the depth of insight into how the OLR method would perform in different settings or with different types of data. Overall, expanding the experimental results by adding more baselines and more datasets can improve the message of the paper.

2. A weakness in the paper is the limited novelty of the OLR approach, especially when considering the prior introduction of a similar regularization concept aimed at achieving permutation invariance from Cohen-Karlik et al., 2020. This earlier work suggests that the core idea of using regularization to mitigate order sensitivity in RNNs has already been explored, potentially diminishing the perceived innovation of the OLR method. Addressing this, a more thorough discussion of how OLR distinguishes itself from previous approaches and contributes new insights or improvements would be beneficial to strengthen the paper's novelty claim.

3. The paper's exclusive focus on Recurrent Neural Networks (RNNs) for graph generation, without discussing the potential applicability or comparison with Transformer models, is a weakness. Given the substantial advances and successes of Transformers in various domains of machine learning, including graph processing and generation, their omission from the discussion limits the breadth of the paper's exploration into modern architectures. This oversight may leave readers questioning whether the OLR method could be adapted to Transformer-based models, which are known for their superior handling of sequences and relational data.

---

> ### Author Response · Authors · 2024-04-24
> **Author response**
>
> Thank you for the comprehensive review and comments. We have made changes in the manuscript to reframe the contribution accordingly. Below we address each point raised.
>
> 1. We address the reviewer’s concern in two separate ways:
> - We have run the molecular generation task on additional architectures: GRU and RNN. The GRU results have been included in the table for comparison, and the RNN results were non-competitive, as described in the text. The outcome is that the effect of OLR on the GRU architecture is similar to that seen in the LSTM architecture. Please see the description in Section 5.2.
> - We have reframed our main contribution, to emphasize that our method has been developed specifically to be applicable to the molecular use case.  This use case is of particular interest as SMILES strings arise naturally as flattened versions of graphs; and due to the fact that there is a thriving interest in developing algorithms on SMILES strings within the ML community (e.g. see the various references cited in the paper, both in the Introduction and the Related Work section).  Changes have been introduced in the title, abstract, and introduction to reflect this reframing.
> 2. We have added a paragraph in the related work which differentiates our work from that of Cohen-Karlik et al. 2020 (Section 4).
> 3. We added a Discussion section where we detail the changes necessary for implementing OLR for the Transformer architecture (Section 6).

---

### Review · Reviewer_77ej · 2024-04-04

**Summary Of Contributions:**

This work aims to design a method that is invariant to the ordering of different DFS trajectories when generating graphs using RNNs.

1. With definitions of several necessary concepts about the ordering of DFS and induced subgraphs, it proposes a regularizer that penalizes the distance between model outputs on different DFS trajectories, which is hopefully to induce a related invariant property.

2. It shows the efficiency of sampling different DFS trajectories with common end vertex on two classes of graphs.

3. The proposed method is experimentally verified on two tasks including molecular generation.

**Audience:**

Yes

**Broader Impact Concerns:**

N/A.

**Claims And Evidence:**

Yes

**Requested Changes:**

Please see the above concerns. All of the four points will have a significant impact on the evaluation.

**Strengths And Weaknesses:**

Pros:
1. The motivation is great: while there have been many discussions about the relationship between permutation symmetry and GNNs for a given graph, it seems interesting and promising to consider invariance to different generation trajectories in the context of graph generation.
2. The proposed regularizer is simple and general: it is independent of specific choice of models, which might be helpful to develop other models with such a property of invariance.
3. The reported numerical results in Table 1,2 are good compared with some baselines.

Cons:
1. As the regularizer is general and independent of specific choices of models, it would be good to evaluate this method on more models. For now, the conducted experiments are a little limited, making the results less convincing. A good example would be Table 1 in [1], where their method is also free of architectures.

2. The definition 3.5 of $k$-edge-connected seems different from what was intended. It needs an additional condition ''there exists $|\widetilde{\mathcal{E}}|=k$ such that the induced $G'$ is disconnected''. It current definition is not correct as any $(k+1)$-edge-connected graph is also $k$-edge-connected for any $k$. Btw, I am not sure, if the new condition is added, whether the original condition is needed any more, since DFS can still be conducted on disconnected graph.

3. In Definition 3.4 and (3.4), more clarification of *totally structure invariant* is needed from two aspects.

    a. For any $\\widetilde{G}\\in\\mathcal{G}_{DFS}(G)$, any vertex $v\\in\\widetilde{G}$ does not necessarily have a DFS trajectory in $G$ that ends with $v$. This seems a little misaligned with the motivation of this work in Def 3.1.

    b. Similar to the previous point, any vertex $v\\in\\widetilde{G}$ does not necessarily have a DFS trajectory in $\\widetilde{G}$ that ends with $v$. This might raise some questions in the implementation.

4. Similar to the above point 3.b, Prop 3.6 is about sampling different trajectories ending at a certain common vertex, but can this kind of sampling be done for any vertex $v$ in subgraph $\\widetilde{G}$?

Reference:

[1] Lim et al. Sign and Basis Invariant Networks for Spectral Graph Representation Learning.

---

> ### Author Response · Authors · 2024-04-24
> **Author response**
>
> We thank the reviewer for the detailed review and suggestions. We address each point raised below.
> 1. We address the reviewer’s concern in two separate ways:
> - We have run the molecular generation task on additional architectures: GRU and RNN.  The GRU results have been included in the table for comparison, and the RNN results were non-competitive, as described in the text.  The outcome is that the effect of OLR on the GRU architecture is similar to that seen in the LSTM architecture. Please see the description in Section 5.2.
> - We have reframed our main contribution, to emphasize that our method has been developed specifically to be applicable to the molecular use case.  This use case is of particular interest as SMILES strings arise naturally as flattened versions of graphs; and due to the fact that there is a thriving interest in developing algorithms on SMILES strings within the ML community (e.g. see the various references cited in the paper, both in the Introduction and the Related Work section).  Changes have been introduced in the title, abstract, and introduction to reflect this reframing.
> 2. The reviewer is correct that in its original phrasing the definition is ambiguous, we fixed the definition to capture the maximal k. Thanks for pointing this out!
> While it is true that DFS can be conducted on disconnected graphs, we focus in this work on the different representations of a given graph. The definition of k-edge connectivity is necessary since our method for generating different representations for a given connected component of a graph relies on the min-cut as described in Proposition 3.6. Our results can easily be carried over to disconnected graphs (by considering each connected component  separately), however we chose to limit the discussion to connected graphs for the sake of clarity.
> 3.
> - The reviewer is correct to point out that not every node in the graph can be a terminal node for a valid DFS trajectory. We changed Definition 2.2. to address this. We would like to emphasize that this is an (important) technical observation which does not have implications on the motivation of  this work. On the contrary, the motivation of this work is based on the intricacies arising from a certain structure of a molecule and the many representations it induces.
> - The implementation is based on the proof sketch of Proposition 3.6 which results only in valid DFS trajectories and therefore cannot generate trajectories with non-terminal nodes.
> 4. The reviewer is correct to point out that not every node in the graph can be a terminal node of some DFS trajectory. We added an explicit remark at the end of Section 3.3. which mentions that our method of generating different trajectories is not comprehensive and there may be additional trajectories that are not explored by our method. However, we prove empirically that our approximation is effective when data is scarce

---

### Decision · Action_Editor_yaeC · 2024-06-17

**Recommendation:** Accept with minor revision

**Comment:**

The proposed approach of regularizing auto-regressive models based on different orderings is interesting, and complemented by thorough empirical evidence.

However, two out of three reviewers raised concerns regarding the experiments, namely on the limited set of datasets considered, and more importantly other models to compare to.

A small note on the Transformer discussion in Section 6, since it is not clear to me why the proposed method doesn't apply to Transformers. In particular, with causally masked attention and positional embeddings, these models are no longer permutation invariant, and if I understand correctly, the regularization applies only to the last hidden layer before the final prediction, which can certainly be done in an auto-regressive Transformer as well?

Please address the above in a minor revision, either by further clarifying why the method doesn't apply to Transformers, or by including some simple experiments on auto-regressive (causal) Transformers. These seem important for the thoroughness of the experiments since they are currently the most widely adopted sequential architecture, and thus an important baseline to include for the TMLR community.

**Audience:**

The proposed approach is of interest to the graph generation community, and is the manuscript is thus suitable for the TMLR audience.

**Claims And Evidence:**

The paper proposes a new regularization strategy for graph generation based on auto-regressive modeling with different orderings. The motivation, claims, and results are supported by clear evidence.

The reviewers raised some concerns regarding the thoroughness of the empirical study, which is currently limited to two variants of RNN architectures.

---

> ### Author Response · Authors · 2024-07-14
> **Revised Manuscript**
>
> We are grateful for the opportunity to contribute to the TMLR community and for the constructive feedback from the Action Editor and reviewers.
>
> In response to the Action Editor’s comments, we have conducted additional experiments with auto-regressive (causal) Transformers and revised the manuscript accordingly. The text in Section 6 has been updated to clarify the applicability of our method to Transformers. These changes are marked in blue in the revised manuscript.

---

> > ### Comment · Action_Editor_yaeC · 2024-07-15
> > **Thank you**
> >
> > Dear authors,
> >
> > Thank you for the changes. Please submit the camera ready revision.
> >
> > AE